# An Adaptive Network Coding Scheme for Multipath Transmission in Cellular-Based Vehicular Networks

**DOI:** 10.3390/s20205902

**Published:** 2020-10-19

**Authors:** Chenyang Yin, Ping Dong, Xiaojiang Du, Tao Zheng, Hongke Zhang, Mohsen Guizani

**Affiliations:** 1School of Electronic and Information Engineering, Beijing Jiaotong University, Beijing 100044, China; 17111009@bjtu.edu.cn (C.Y.); zhengtao@bjtu.edu.cn (T.Z.); hkzhang@bjtu.edu.cn (H.Z.); 2Department of Computer and Information Sciences, Temple University, Philadelphia, PA 19122, USA; dxj@ieee.org; 3Department of Computer Science and Engineering, Qatar University, Doha 2713, Qatar; mguizani@ieee.org

**Keywords:** vehicular network, multipath transmission, network coding, machine learning

## Abstract

With the emergence of vehicular Internet-of-Things (IoT) applications, it is a significant challenge for vehicular IoT systems to obtain higher throughput in vehicle-to-cloud multipath transmission. Network Coding (NC) has been recognized as a promising paradigm for improving vehicular wireless network throughput by reducing packet loss in transmission. However, existing researches on NC do not consider the influence of the rapid quality change of wireless links on NC schemes, which poses a great challenge to dynamically adjust the coding rate according to the variation of link quality in vehicle-to-cloud multipath transmission in order to avoid consuming unnecessary bandwidth resources and to increase network throughput. Therefore, we propose an Adaptive Network Coding (ANC) scheme brought by the novel integration of the Hidden Markov Model (HMM) into the NC scheme to efficiently adjust the coding rate according to the estimated packet loss rate (PLR). The ANC scheme conquers the rapid change of wireless link quality to obtain the utmost throughput and reduce the packet loss in transmission. In terms of the throughput performance, the simulations and real experiment results show that the ANC scheme outperforms state-of-the-art NC schemes for vehicular wireless multipath transmission in vehicular IoT systems.

## 1. Introduction

The future vehicular Internet-of-Things (IoT) is an important branch of IoT. The development of vehicular IoT has promoted the development of vehicular applications. In an intelligent transportation system (ITS), novel vehicular applications, such as cooperative autonomous driving, federated learning, blockchain, Virtual Reality (VR), and Augmented Reality (AR), which are sensitive to transmission delay, bandwidth, and throughput, are emerging [1,2,3,4]. The data that are generated by these new applications need to be transmitted to the cloud servers through vehicle-to-cloud communication [5]. A major bottleneck of vehicle-to-cloud communication is the limited bandwidth and poor communication quality in vehicular wireless networks [6]. To improve the transmission bandwidth of the vehicular network, Duo et al. [7] proposed introducing cellular-based wireless networks (e.g., 5G, 4G, etc.) into the vehicular ad hoc network (VANET), Mandala et al. [8] proposed even energy dissipation protocol (EEDP) to forward sensor data to the base station to balance the traffic load in sensor network, and Brown and Du [9] described an efficient scheme for reporting packet drops in the transmission from sensor node to the base station. However, the bandwidth that is provided by a single cellular wireless link is unable to meet the bandwidth requirements of these novel applications in vehicular IoT. In recent years, many papers [10,11,12,13,14] proposed different multipath transmission algorithms to transmit data packets over multiple links to aggregate bandwidth. The multipath transmission has been proven to be a powerful method for increasing the bandwidth of vehicle-to-cloud communication system by aggregating available cellular wireless resources. Dong et al. [15] proposed to introduce the multipath transmission into the vehicle-to-cloud communication to improve the transmission quality. Zhang et al. [16] proposed the Receiver Adaptive Incremental Delay (RAID) scheme for vehicle-to-ground multipath communication to mitigate the impact of packets disorder on network throughput. However, the throughput performance of multipath transmission in the vehicular wireless network is seriously affected by packet loss. Li et al. [17] pointed out that when the packet loss rate (PLR) of the wireless link is getting higher, the throughput will decrease significantly because there are too many data packets lost in transmission. There are many researchers dedicated to the packet loss problem in a NC perspective. These works are divided into two categories. One type is NC schemes based on repetition code, Opportunistic Routing (OR) is one of them [18,19]. OR is proposed to avoid packet loss by sending a copy of the packet through each available wireless link. The other type is NC schemes based on erasure code (EC), which could recover lost data by adding redundant data, refs. [20,21,22,23] realized the NC schemes based on EC by different methods, such as bitwise XOR, Galois Field, Big Number, etc. The coding rate of the NC scheme based on EC represents the ability to overcome packet loss in transmission. The coding rate is the ratio of the number of redundant packets *r* to the total number of coded packets *n*, which could be represented as r/n.

However, these NC schemes do not consider the effect of link quality variations in coding and decoding processes. The link quality of the wireless link is changing at all times in reality due to the influences of vehicle movement, obstacles, and other factors. The rapid change of link quality reflects in the variations of PLR in the wireless link. The existing NC schemes cannot dynamically adjust the coding rate according to the variations of PLR in the wireless link, which could lead to two serious problems. First, when the quality of the wireless link becomes better, the PLR of the wireless link becomes lower from the high level. Although schemes, like BigNum Network Coding (BNNC) [23] scheme and Galois Field Network Coding (GFNC) [22], can ensure there is no packet loss, they consume a lot of unnecessary bandwidth because these schemes still use a relatively high coding rate to ensure network reliability. A high coding rate results in low bandwidth utilization. Accordingly, the throughput performance of the vehicular wireless network is poor. Second, in a contrary case, when the quality of wireless link declines, the PLR of the wireless link becomes a higher level from the low level, the above-mentioned NC schemes with relatively low coding rate cannot cope with the poor quality of the wireless link, which means that the receiver fails to decode coded packets, because there are not enough redundant packets to cope with the packet loss in transmission. To summarize, the performance of these NC schemes is poor for the variations of link quality.

In this paper, aiming to counter the rapid change of wireless link quality, we investigate how to optimize network coding to dynamically adjust the coding rate in order to increase vehicular-to-cloud transmission throughput and reduce packet loss. Especially, to support real-time coding rate adjustment, we design the architecture of the ANC scheme for vehicle-to-cloud multipath transmission. Subsequently, based on the designed architecture, a novel network coding scheme is introduced to dynamically adjust the coding rate according to the estimated next moment PLR to increase throughput and reduce packet loss. Finally, we evaluate its performance when compared to state-of-the-art NC schemes for vehicular multipath transmission, including the OR scheme, BNNC scheme, and GFNC scheme. In terms of throughput and packet loss, extensive simulations and real experiments show that ANC can greatly improve throughput performance and effectively reduce packet loss in transmission.

Our main contributions in this paper are summarized as follows:We propose an adaptive network coding scheme for vehicle-to-cloud multipath communication in cellular-based wireless networks. When compared with the current NC schemes, ANC significantly improves network throughput performance and reduces multipath transmission packet loss.We propose a novel combination of network coding and packet dispatch to reduce packet caching time. Packet dispatch is realized in a group aspect to determine the distribution proportion in each available link according to measured bandwidth and Round-Trip Time (RTT).We introduce wavelets into Hidden Markov Model (HMM) to fit in with the rapid change of link quality in the cellular-based vehicular wireless networks. This estimation method effectively reduces the error range of estimated PLR.

The structure of the paper is organized, as follows. In Section 2, some related works about multipath transmission and NC schemes are introduced. In Section 3, we give an overview of the ANC scheme. In Section 4, we introduce every part of the ANC scheme in detail. In Section 5, we take lots of simulations and real experiments in order to prove that our scheme is effective and reliable. Finally, Section 6 provides concluding remarks of this work.

## 2. Related Work

### 2.1. Multipath Transmission Schemes in Cellular-Based Vehicular Networks

Many multipath transmission schemes in cellular-based vehicular networks have been proposed in the perspectives of the transport layer and the IP layer. On the transport layer, SCTP and MPTCP are mainstream multipath transmission schemes. Xu et al. [24] proposed a novel cross-layer fairness-driven (CL/FD) SCTP-based concurrent multipath transfer (CMT) solution (CMT-CL/FD) in order to improve video delivery performance. Chung-Ming Huang and Ming-Sian Lin [25] proposed to have a loss detection mechanism in SCTP (RG-SCTP) for vehicular networks to reduce the influence of packet loss in transmission between On Bus Unit (OBU) and Road Side Unit (RSU). There are also many researches regarding MPTCP, NC-MPTCP [13], and QCBF-MPTCP [26] worked on MPTCP to improve the throughput performance by collision avoidance and sub-flow scheduling. When compared with the transport layer multipath transmission, IP layer multipath transmissions do not need to design new protocol, they are easier to be deployed and implemented. Generally, researchers realize IP layer multipath transmission with the method of packet encapsulation. IP-in-IP is a kind of packet encapsulation that attaches a new IP header with the original IP packet in order to form a new packet. Based on this approach, Dong et al. [27] proposed a multipath transparent transmission scheme on the IP layer in vehicular wireless networks. This scheme can make full use of heterogeneous wireless networks between ground-based servers and onboard smart devices by building a virtual tunnel. Based on [27], Zhang et al. [16] proposed RAID algorithm to overcome out-of-order data packets in heterogeneous wireless networks.

There are also a lot of researches on multipath transmission combined with artificial intelligence. Xu et al. [28] designed a deep reinforcement learning (DRL)-based control framework DRL-CC for Congestion Control. Naeem et al. [29] proposed a novel model-free SDN-based adaptive deep reinforcement learning framework that was based on a fuzzy normalized neural network to address the issue of congestion control for MPTCP in the IoT networks. Arianpoo et al. [30] proposed a combination of SCTP and Q-learning to solve the receiver buffer blocking problem. However, there are not many researches on IP layer multipath transmission combined with artificial intelligence.

### 2.2. NC Researches

NC was proposed by [31] to reach the upper limit of Network multicast capacity. In recent years, NC has been applied to multipath transmission. Many researches have been proposed. Kim et al. [32] combined OR with multipath transmission to improve wireless sensor network reliability. Simply speaking, the essence of OR is repetition code, OR sends data packets to the next sensor node through all available wireless links. Assuming that the packet loss probability of each link is *p*, the overall PLR of *n* links is $p^{n}$, it can effectively reduce the packet loss probability of the entire system. However, the drawback of this idea is obvious. In terms of resource utilization efficiency, for a system with *n* communication links, each time a sent packet is received by the next sensor node, it means that n−1 redundant packets need to be simultaneously discarded. Lin et al. [33] proposed SlideOR to encode raw packets in overlapping sliding windows. Xu et al. [34] proposed MPTCP-PNC to address packet reordering problem. When considering the scheduling of coded packets in each available link, Xu et al. [22] proposed combining SCTP with NC. To raise the anti-jamming capability of transmission, Zhang et al. [23] proposed a Big Number network coding scheme to reduce packet loss during transmission.

## 3. ANC Scheme Overview

Figure 1 is the architecture of the ANC scheme, which consists of three parts: sender, receiver, and multiple wireless links. The ANC scheme is deployed in sender and receiver. Through the cooperation between the modules that are embedded in sender and receiver, the real-time information, such as estimated PLR, measured bandwidth, and RTT, can be used in order to determine the coding rate to change the ability to overcome packet loss in the transmission according to the variation of wireless link quality.

The sender includes ANC Mapping Module, ANC Coding Module, Sender Buffer and PLR Estimation Module. The PLR Estimation Module connects with Sender Buffer to obtain link quality during transmission. PLR Estimation Module estimates next moment PLR. It includes PLR data acquisition and data estimation, which is introduced in detail in Section 4.4. First, the ANC Mapping Module obtains data from vehicular applications and divides packets into several groups. The number of groups is determined by the number of available wireless links. Next, according to real-time information about link quality, ANC Mapping Module determines the number of raw packets and the number of redundant packets for a group and delivers information to ANC Coding Module. The ANC Coding Module uses estimation and measurement information to generate different coding matrices corresponding to different groups. Subsequently, ANC Coding Module encodes the raw packets within a group to form a coded group. Sender Buffer sends coded groups via different wireless links.

The process of multipath transmission between the sender and receiver can be concluded into two steps. First, the sender encapsulates the coded IP data packet into the data part of the IP packet and attaches a new header to form a new IP packet. Afterwards, the sender transmits these packets to the receiver through different wireless links simultaneously.

The receiver uses ANC Mapping Module to remove the customized IP header to obtain the information about the decoding process. Subsequently, the ANC Mapping Module converts processed packets into numbers and store them in computer memory. When the receiver obtains enough packets to be decoded, the receiver uses ANC Decoding Module to decode coded groups according to the information about the decoding process, and store raw packets in Receiver Buffer. Receiver Buffer is not only responsible for storing the raw packets, but also responsible for forwarding raw packets to the application servers.

## 4. The Detail of ANC Scheme

In this section, we introduce the ANC scheme in detail. The structure of this section is organized as follows. First, We give definitions of coding rate and transmission failure rate (TFR) in Section 4.1. Next, we introduce the network topology model in a typical vehicular network scenario in Section 4.2. Subsequently, we introduce the coding and decoding processes of the ANC scheme in Section 4.3. Finally, Section 4.4 provides the mathematical PLR estimation Model.

### 4.1. Preliminaries and Definition

Before we introduce the detail of the ANC scheme, there are two terminologies that we must explain:Coding rate: the coding rate is the ratio of the number of redundant packets to the total number of coded packets. For example, if *k* raw packets are encoded into *n* coded packets, then, there are r=n−k redundant packets generated to combat the packet loss of wireless links. In this case, the coding rate is r/n. When r/n becomes higher, it means NC generates more redundant packets. When r/n becomes lower, it means that NC generates fewer redundant packets.TFR: TFR is the ratio of the number of coded groups that fail to be decoded to the total number of the coded groups. Although NC can reduce the probability of packet loss, it can not completely solve the problem of packet loss in wireless transmission. There are still some packets lost in transmission. When there are too many lost packets in transmission, the receiver can not obtain enough coded packets to support the decoding process. TFR can well quantify the ability of the NC scheme in order to overcome the packet loss problem of a wireless link in the different communication environments. Let Failnum denote the number of the coded group that fails to be decoded. We use Transnum to represent the total number of groups transmitted over a certain link. Accordingly, TFR can be calculated as Failnum/Transnum.

The coding rate is first mentioned in Section 2 and it will be used repeatedly in the rest of this paper. TFR is the abbreviation of the transmission failure rate that is first mentioned in Section 4.1. TFR is a vital index for evaluating the ability of the NC scheme to cope with the packet loss in the vehicular wireless network, which will be used in Section 5.

### 4.2. Network Topology Model

Figure 2 is a vehicle-to-could multipath transmission network topology. The vehicle-to-could multipath transmission system is to combine vehicular cyber-physical systems with cloud computing technologies to offer essential services for passengers and drivers. The architecture includes three layers: on-board cyber-physical system, cellular-based wireless networks, and cloud platform system. On-board cyber-physical system consists of a large number of on-board IoT devices and a mobile access router (MAR), as shown in Figure 2. MAR is equipped with several wireless interfaces to connect with *M* heterogeneous cellular wireless networks at the same time. *M* generally is 3 indicating the three major Internet service providers (ISP) in China currently. On-board IoT devices, such as smartphones and laptops, are connected with a MAR via Wi-Fi or data cable. On-board IoT devices transmit data that include real-time record video and train running state to the cloud server via MAR. Subsequently, MAR transmits data to the cloud through different cellular-based wireless links at the same time. As shown in Figure 2, the cloud platform system includes the mobile edge router (MER) and cloud servers. MER is a network aggregation terminal device deployed in the cloud, which is used to aggregate coded data packets that are sent from different wireless links by MAR and decode the coded packets to recover lost raw packets in transmission. Subsequently, MER forwards raw data packets to different application servers. The ANC scheme is both deployed in MAR and MER.

### 4.3. The Coding and Decoding Processes of ANC

Before we introduce the coding and decoding process of the ANC scheme in detail, we provide Table 1 in order to illustrate the meaning of the abbreviations mentioned below.

In the entire encoding process, the ANC Mapping Module is responsible for dividing data packets into different groups and determine the number of redundant packets. First, ANC Mapping Module decides the number of raw packets within a group with the measured bandwidth and RTT. Let $BW_{i}$ denote the bandwidth of the *i* th wireless link. $RTT_{i}$ represents the RTT of the *i* th wireless link. To simplify the problem, we regard the length of the packet as the Maximum Transmission Unit (MTU) and we suppose the $BW_{i}$ and $RTT_{i}$ are fixed when the ANC scheme adjusts the coding rate. Accordingly, we can define the total delay for one packet in transmission according to the measurement of RTT and bandwidth.
(1)$$\Delta T_{i}=MTU/BW_{i}+RTT_{i}/2.$$

Equation (Equation 1) consists of two parts: $MTU/BW_{i}$ and $RTT_{i}/2$. $MTU/BW_{i}$ represents the sending delay of one packet. $RTT_{i}$ consists of three parts: the propagation delay of the *i* th wireless link, the queuing delay in the routers and end devices, and the processing delay in the router cache. Accordingly, $RTT_{i}/2$ represents the sum of the propagation delay, queue delay, and processing delay of one packet in transmission. In the Equation (Equation 2), $\Theta_{i+1}$ represents the number of packets that link i+1 can transmit in the time link *i* takes for one packet to transmit to the receiver.
(2)$$\Theta_{i+1}=\Delta T_{i}/\Delta T_{i+1}$$

Therefore, we can propose an algorithm to dynamically determine the proportion of coded packets allocated on each available wireless link.

Let $n_{i}$ represents the number of coded packets within a coded group E in the *i* th wireless link, which is the same as the number of row vectors of CM. Let $k_{i}$ represent the number of raw packets in a group B before coding in the *i* th wireless link, which is the same as the number of column vectors of CM. Let $r_{i}$ represent the number of redundant packets in a coded group E in the *i* th wireless link, which is the same as the number of row vectors of RM. Let $p_{i}^{raw}$ represent the packet loss rate of the data packet that is not processed by any network coding scheme during transmission on the *i* th wireless link. Hence, $n_{i}=k_{i}+r_{i}$. The number of successfully received coded packets in E after transmission can be calculated as $\left(1-p_{i}^{raw}\right)n_{i}$. We can obtain the relationship between $\left(1-p_{i}^{raw}\right)n_{i}$ and $k_{i}$ when considering that the receiver need obtain more than $k_{i}$ coded packets to recover raw packets.
(3)$$r_{i}^{min}\geq Ceil\left(k_{i}\frac{p_{i}^{raw}}{1-p_{i}^{raw}}\right).$$

Ceil() is a function that always rounds a number up to the next largest integer. $r_{i}^{min}$ is the minimum number of redundant packets in the *i* th wireless link that ensures the receiver can recover all raw packets. ANC Mapping Module makes inequality (Equation 3) be used in conjunction with Algorithm 1 to determine the number of coded packets ($k_{i}$) in the *i* th wireless link. The combination makes coded packets in each available link have the ability to reduce the packet loss of the wireless link. Let *W* denote the total number of data packets that are obtained by ANC Mapping Module from different applications. To simplify equations, we suppose there are 3 available wireless links, and the packets distribution proportion of link 1 is the minimum. Therefore, $k_{1}+k_{2}+k_{3}=W$ and $\Theta_{1}=1$ according to Algorithm 1. $k_{i}\left(1+\frac{p_{i}}{1-p_{i}}\right)$ represents the number of coded packet. Therefore, we proposed a system of linear equations for packet dispatch, as below,
(4)$$\left\{\begin{array}{c} k_{1}+k_{2}+k_{3}=W\\ k_{1}\left(1+\frac{p_{1}^{raw}}{1-p_{1}^{raw}}\right)\Theta_{2}=k_{2}\left(1+\frac{p_{2}^{raw}}{1-p_{2}^{raw}}\right)\\ k_{1}\left(1+\frac{p_{1}^{raw}}{1-p_{1}^{raw}}\right)\Theta_{3}=k_{3}\left(1+\frac{p_{3}^{raw}}{1-p_{3}^{raw}}\right) \end{array}\right.$$

$k_{1},k_{2},k_{3}$ could be obtained by solving Equation (Equation 4). The number of redundant packets $r_{i}$ is equal to $k_{i}\frac{p_{i}^{raw}}{1-p_{i}^{raw}}$. In order to generalize Equation (Equation 4), let $\Theta_{min}$ denote the minimum of Θ and $i_{min}$ denote the link with $\Theta_{min}$ distribution proportion. The number of available links is *N*. We define $i\in N/\Theta_{i}\neq \Theta_{min}$ to represent every available link, except the link whose packets distribution proportion is the minimum. The system of linear equations is concluded, as below,
(5)$$\left\{\begin{array}{cc} \sum_{i=1}^{N}k_{i}=W\\ k_{i_{min}}\left(1+\frac{p_{i_{min}}^{raw}}{1-p_{i_{min}}^{raw}}\right)\Theta_{i}=k_{i}\left(1+\frac{p_{i}^{raw}}{1-p_{i}^{raw}}\right) & \forall i\in N/\Theta_{i}\neq \Theta_{min} \end{array}\right.$$

The number of raw packets in the *i* th link can be obtained with the linear equations system (Equation 5). Subsequently, $r_{i}$ is $Ceil(k_{i}\frac{p_{i}^{raw}}{1-p_{i}^{raw}})$. ANC Mapping Module transmits the information including $k_{i}$ and $r_{i}$ of the *i* th link to the ANC Coding Module. According to $k_{i}$ and $r_{i}$, the ANC Coding Module determines the row number and column number of ${\mathrm{CM}}_{i}$. We only consider one wireless link to simplify the description. Therefore, ${\mathrm{CM}}_{i}$ is replaced by CM and $k_{i},n_{i},r_{i}$ is replaced by k,n,r. Each group has *k* raw packets. *k* is different in each available link. ANC Coding Module regards each group as a column vector,
(6)$$B={\left[\begin{array}{cccc} b_{1} & b_{2} & \cdots & b_{k} \end{array}\right]}^{T}\left(k\times 1\right).$$

**Algorithm 1** Coded packets distribution
**Input:** measured RTT, measured BW, estimated next moment PLR
**Output:** packets distribution proportion Θ
1:This algorithm is used to determine the proportion of packets allocated on each available wireless link2:ΔT=[];Δmax=0;Θ=[];3:**for** each available Path *i*
**do**4:    $\Delta T_{i}=\frac{1}{BW_{i}}+\frac{RTT_{i}}{2}$;5:    $\Delta T.append(\Delta T_{i})$;6:
**end for**
7:**for***j* in ΔT
**do**8:    **if**
j≥Δtemp
**then**9:        Δmax=j;10:    **end if**11:
**end for**
12:**for**k=0 to len(ΔT)−1
**do**13:    $\Theta_{k}=\frac{\Delta max}{\Delta T_{k}}$;14:    $\Theta .append(\Theta_{k})$;15:
**end for**
16:
Ceil(Θ)
17:
**return**
Θ



CM is coding matrix that consists of *k* rows identity matrix (IM) and n−k rows linearly independent redundant matrix (RM),
(7)$$\mathrm{CM}=\left[\begin{array}{c} IM_{k\times k}\\ RM_{(n-k)\times k} \end{array}\right]={\left[\begin{array}{cccc} 1 & 0 & \cdots & 0\\ 0 & 1 & \cdots & 0\\ \vdots & \vdots & \ddots & \vdots \\ 0 & 0 & \cdots & 1\\ a_{k+1,1} & a_{k+1,2} & \cdots & a_{k+1,k}\\ a_{k+2,1} & a_{k+2,2} & \cdots & a_{k+2,k}\\ \vdots & \vdots & \ddots & \vdots \\ a_{n,1} & a_{n,2} & \cdots & a_{n,k} \end{array}\right]}_{n\times k}.$$

RM and CM have *k* column vectors, *k* is the number of raw packets. *n* is the total row number of CM. n−k is the number of redundant packets. RM from CM is composed of (n−k)×k Vandermonde matrix, Vandermonde matrix is represented, as follows,
(8)$$\mathrm{RM}={\left[\begin{array}{cccc} 1 & 1 & \cdots & 1\\ a_{k+2,1} & a_{k+2,2} & \cdots & a_{k+2,k}\\ a_{k+2,1}^{2} & a_{k+2,2}^{2} & \cdots & a_{k+2,k}^{2}\\ a_{k+2,1}^{3} & a_{k+2,2}^{3} & \cdots & a_{k+2,k}^{3}\\ \vdots & \vdots & \ddots & \vdots \\ a_{k+2,1}^{n-k-1} & a_{k+2,2}^{n-k-1} & \cdots & a_{k+2,k}^{n-k-1} \end{array}\right]}_{(n-k)\times k}.$$

Let $a_{k+2,1},a_{k+2,2},\ldots ,a_{k+2,k}$ be 1,2,…,k. The process of coding is linear operation shown in Formula (Equation 9), multiply CM and B to obtain a group of coded packets E. The length of column vector E is *n*.
(9)$$E_{n\times 1}={\mathrm{CM}}_{n\times k}\times B_{k\times 1}\left(n\geq k\right).$$

To illustrate the process of coding in a more intuitive perspective, we explain Formula (Equation 9) by the following Figure 3. In Figure 3, CM is a coefficient matrix of the system of linear equations. There are five packets divided into one group. The coding operation is the multiplication of CM and group B. Because CM consists of *k* rows IM and n−k rows RM, group E consists of raw group packets B and redundant packets ($C_{1},C_{2},C_{3}$) generated by RM. Some packets in group E could be lost in multipath transmission. In Figure 3, $B_{1}$, $B_{4}$, and $C_{1}$ packets are lost.

Due to the packet loss in transmission, receiver can not get the complete coded packets group E. Therefore, E is converted into $\hat{E}$. $\hat{E}$ is formed by packets that are not lost in transmission. In Figure 3, $\hat{E}$ includes $B_{2}$, $B_{3}$, $B_{5}$, $C_{2}$, and $C_{3}$ packets.

The IP header of received coded packet has a 6-tuple θ=(MSN,RPL,CPL,SSN,GN,LN), which is introduced in Figure 4. MSN is used in order to determine the row vector sequence number of the encoding matrix corresponding to the encoded packet, CPL is used to confirm the total number of coded data packets within a group and RPL is used to conform the number of redundant packets within a group. We use CPL to confirm the number of row vectors in matrix CM and use CPL−RPL to confirm the number of column vectors in matrix CM. Because the coefficients of RM is defined as 1,2,…,k, in receiver, ANC can confirm every coefficients in RM according to the CPL and CPL−RPL. Because of the packets loss in transmission, CM is converted into $\hat{\mathrm{CM}}$. $\hat{\mathrm{CM}}$ includes a certain number of row vectors belonging to RM and a certain number of row vectors belonging to IM. In the receiver, ANC can determine the row vector in $\hat{\mathrm{CM}}$ according to the MSN and RM. For example, in Figure 3, $\hat{\mathrm{CM}}$ consists of the row vectors that are corresponding to $B_{2}$, $B_{3}$, $B_{5}$, $C_{2}$ and $C_{3}$. Due to row vectors in $\hat{\mathrm{CM}}$ are linearly independent by each other, $\hat{\mathrm{CM}}$ is invertible.

We conclude the core idea of decoding in Formula (Equation 10). As long as any *k* packets in the vector E are received, we could obtain $\hat{E}$. Because $\hat{\mathrm{CM}}$ is invertible, D could be calculated as an inverse matrix of $\hat{\mathrm{CM}}$. We usually call D the decoding matrix. $\hat{E}$ can be multiplied with the decoding matrix D to recover the original *k* raw packets.
(10)$$B_{k\times 1}=D_{k\times k}\times {\hat{E}}_{k\times 1}.$$

### 4.4. Mathematical PLR Estimation Model

We need to solve the system of linear Equation (Equation 4) to determine the number of raw packets *k* and the number of redundant packets *r* in order to dynamically adjust the coding rate to counter the variation of the wireless link. However, the coefficient $p_{i}^{raw}$ is a random variable, we need a mathematical model to estimate the next moment raw PLR to confirm $p_{i}^{raw}$. In this subsection, we introduce the detail of next moment raw PLR estimation, which includes two steps: average PLR acquisition and raw PLR estimation. In data acquisition, we use MAR to transmit packets that are not processed by any network coding schemes with the customized header through a single vehicular wireless link over a period of time to calculate the average raw PLRs of a single link. In PLR estimation, we use WDD to process the average raw PLR in order to mitigate the estimation interference caused by the violent quality fluctuation of the vehicular wireless network on a small time scale. ANC regards raw PLR as a continuous random variable. Based on the average raw PLRs, ANC could obtain the probability density function (PDF) of raw PLR and estimate the next moment raw PLR with the method of HMM. The estimated next moment raw PLR is used in the system of linear Equation (Equation 4) to dynamically adjust the coding rate.

#### 4.4.1. Average PLR Acquisition

We first introduce the process of data acquisition. The average raw PLR is calculated with the information stored in the customized header. The customized header is shown in Figure 4. Before the coding process, MAR would encapsulate the customized header and probe IP packet into the data part of the new IP data packet. After that, MAR sends the new IP data packet to MER to obtain the statistics of raw PLR. The customized header includes a six-tuple θ=(MSN,RPL,CPL,SSN,GN,LN). MSN is used to determine the row vector sequence number of the encoding matrix, RPL is used to confirm the number of redundant packets within a group, CPL is used to confirm the total number of coded data packets within a group, SSN is the sequence number of successfully transmitted coded IP data packet in a group, GN is used to confirm which group coded packet belongs to, and LN is used to confirm the wireless link from which coded IP data packet is transmitted. When we calculate average PLR, we only send coded packets over a single link. Accordingly, we use four parameters SSN,CPL,GN,LN mentioned above to calculate the average PLR of each wireless link through the following Algorithm 2.
**Algorithm 2** Average PLR acquisition**Input:** GN, LN, SSN, CPL**Output:** averPLRset1:This algorithm is used to calculate average packet loss rate over time2:averPLRset=[];3:**for** each available Path *i*
**do**4:    i.sendnum=0;i.recvnum=0;i.group=[];i.averPLR=0;5:    **for** each received packet *j*
**do**6:        **if**
j.LN==*i*
**then**7:           i.recvnum++;8:           **if**
i.group=[] or j.GN
not
in
i.group
**then**9:               i.sendnum+=j.CPL;10:               i.group.append(j.GN);11:           **end if**12:        **end if**13:    **end for**14:**end for**15:**for** each available Path *i*
**do**16:    i.averPLR=(i.sendnum−i.recvnum)/i.sendnum17:    averPLRset.append(i.averPLR)18:**end for**19:**return**averPLRset

#### 4.4.2. Raw PLR Estimation

Reference [16] indicates that the network quality fluctuations have been more fierce in the high-speed scenario for the vehicular wireless networks. Many estimation results of network quality are affected by the uncertain violent fluctuation of the vehicular wireless network. To obtain a better PLR estimation, we make a pre-processing after we get calculated average raw PLR with Algorithm 2. We use Wavelet Domain Denoising (WDD) to process the average raw PLRs because WDD is a useful transform analysis method to denoise irregular fluctuations in signals and retain signal characteristics. WDD includes three steps: first, process the noisy signal with wavelet transform; second, deal with the wavelet coefficients to remove the noise; and third, carry out inverse wavelet transform to get the denoised signals. We choose the Daubechies8 wavelet as the wavelet basis function used in WDD. Figure 5a shows the average raw PLR and the denoised average raw PLR preprocessed by WDD, respectively. Figure 5b is the comparison of the average raw PLR and the denoised average raw PLR preprocessed by WDD, we can find the processed average raw PLRs ignore extreme shifts that caused by the violent quality fluctuation of the vehicular wireless network on a small time scale. These extreme shifts could cause interference to the next moment raw PLR estimation. The processed average raw PLR still retains the original characteristics, and the processed average raw PLR reflects the overall trend of PLR. In Section 4.4.2, we will compare the PLR estimation error range of the WDD+HMM method with that of the HMM method to prove the rationality and validity of the WDD+HMM method.

After introducing the data acquisition and data pre-processing, we talk about the HMM of PLR estimation. We propose an HMM that includes a latent variable and an observed variable. The latent variable is a discrete random variable and it represents the quality of vehicular wireless networks. The states of the latent variable are defined as S={G,N,P}, in which G means Good latent state, P means Poor latent state, and N means Neutral latent state. In order to simplify the mathematical derivation and calculation, we make $S_{1}$ represent Good, $S_{2}$ represent Neutral and $S_{3}$ represent Poor. The observed variable is a continuous random variable and it represents the PDF of raw PLR. The top two layers in Figure 6 are a graphical depiction of the HMM for the relationship between the latent variable and the observed variable.

Solid lines with arrow in the hidden layer represent the state transition of the latent variable. Let $\alpha_{ij}$ denote the state transition conditional probability. Given that the state of the latent variable is *i* at time *t*, the probability that the state of the latent variable is *j* at time t+1 equals $\alpha_{ij}$.
(11)$$\alpha_{ij}=p\left(s_{t+1}=j|s_{t}=i\right),i,j\in S.$$

$A=\{\alpha_{ij}$} is the matrix of state transition conditional probabilities, where $\alpha_{ij}$ has been defined in Equation (Equation 11). The latent variable has 3 states, so the matrix size of *A* is 3×3.

Dash lines with arrow between the hidden layer and observed layer describe the emission of the observed variable. Because the observed variable is not a discrete random variable, the emit probability matrix does not exist. Instead, the conditional probability density function of the observed variable can be defined directly, which follows one-dimensional Gaussian distribution N(σ,μ).
(12)$$f\left(x|S;\sigma ,\mu \right)=\frac{1}{2\pi^{\frac{p}{2}}\left|\begin{array}{c} \sigma \end{array}\right|}EXP\left\{-\frac{1}{2}{\left(x-\mu \right)}^{T}\sigma^{-1}\left(x-\mu \right)\right\}$$

The Gaussian distribution of the observed variable corresponding to different latent variable states $S_{i}$ is given as,
(13)$$p^{raw}∼N_{i}\left(\sigma_{i},\mu_{i}|S_{i}\right),i=1,2,3,N_{i}\in N,S_{i}\in S.$$

Parameter $\mu_{i}$ represents the mean of the observed variable corresponding to the latent variable state $S_{i}$, note that, $\mu_{1}\leq \mu_{2}\leq \mu_{3}$. Initially, there is an initial probability distribution that is given to describe the state selection of the latent variable. Where,
(14)$$\Pi =(\Pi_{1},\Pi_{2},\Pi_{3}).$$

HMM is usually defined as a three-tuple λ=(A,N,Π), where:A={$\alpha_{i,j}$} is the matrix of state transition conditional probability, where $\alpha_{ij}$ has been defined in Equation (Equation 11).N={$N_{i}$} is the set of PDF of the observed variable corresponding to different latent variable states, where $N_{i}$ has been defined in Equation (Equation 13).$\Pi =(\Pi_{1},\Pi_{2},\Pi_{3})$ are the initial state probability distribution, $\Pi_{i}$ is the probability that the system starts from latent variable state $S_{i}$.

To illustrate this model in a more intuitive perspective, we put this model in a high-speed railway (HSR) network scenario. The link quality of cellular wireless link dynamic changes with the movement of the train. As the bottom layer in Figure 6 depicts, BS1 and BS2 are two Base Stations (BSes) of the same ISP, they represent different network link choices for the network adapter interface of MAR. When the train is close enough to BS1, but away from BS2, the adapter interface is connected with BS1, network quality performs Good due to the shortest distance between train and BS1, so we use latent variable state $S_{1}$ to represent this link quality state. When the train is in a position where is nearly the same distance from the two BSes, network quality performs Neutral, because the distance between the train and BS1 is becoming large, but the distance between the train and BS2 is smaller, some packets are dropped caused by the handoff between BS1 and BS2, so, we use latent variable state $S_{2}$ to represent this link quality situation. When the train is far away from two BSes, it’s hard to transfer data packets to the two BSes, network quality performs Poor, which is represented by state $S_{3}$. The state transition of the latent variable takes place at a large time scale and it cannot be directly observed. The observable raw PLR variable changes on a relatively small time scale and it follows different probability distributions that correspond to different latent variable states.

We propose Algorithm 3 to estimate the next moment raw PLR. Algorithm 3 can be summarized into the following three steps:Model learning: we use 10 observed raw PLR points to train this model by Baum-Welch algorithm.Prediction: we estimate the classification of 10 raw PLR points by the Viterbi algorithm and calculate the expectation of next moment raw PLR according to the state transition probability matrix and the conditional probability density function of the observed variable.Update: we combine the estimated raw PLR point with the previous nine data points into a new training set, go to step 1 until the data set runs out.
**Algorithm 3** Next moment raw PLR estimation**Input:** average raw PLRs**Output:** estimated next moment raw PLRs1:This algorithm is used to estimate next moment packet loss rate2:$N_{l}$ means the Gaussian distribution corresponding to the *l* th latent variable3:TrainingTimes=*y*; TrainingSetNum=*x*; LatentVariableStateNum=3;4:**for** each available Path *i*
**do**5:    i.TrainingSet=[];6:**end for**7:**for**j=0 to TrainingSetNum−1
**do**8:    **for** each available Path *i*
**do**9:        averPLR=averagePLRacquisition()[i];10:        i.TrainingSet.append(averPLR);11:    **end for**12:**end for**13:estimationPLRset=[];14:**for** each available Path *i*
**do**15:    i.TrainingSet=wavelet(i.TrainingSet)16:    InitialAssignment(A,N,Π);17:    i.estimationPLR=0;Index=0;LatentVariable=[];18:    **for**
k=0 to TrainingTimes−1
**do**19:        (A,N,Π)=Baum−Welch(A,N,Π);20:    **end for**21:    LatentVariable=Viterbi(A,N,Π);22:    Index=LatentVariable[TrainingSetNum−1];23:    **for**
l=0 to LatentVariableStateNum−1
**do**24:        $i.estimationPLR+=A\left[Index\right]\left[l\right]*N_{l}.mean\left(\right)$;25:    **end for**26:    estimationPLRset.append(i.estimationPLR);27:**end for**28:**return**estimationPLRset

We define the real time consumption of once HMM training with *x* points for a single wireless link is f(x). Some samples of f(x) are listed in Table 2. According to the network topology described in Section 4.2, there are 3 wireless links in the vehicle-to-cloud multipath transmission. Therefore, we need to train the HMM for 3 wireless links, respectively. We define the training times for each link is *y*. Accordingly, the total real time consumption of *y* times HMM training with *x* points for 3 wireless links is 3∗f(x)∗y. The packet sending interval of NS-3 in simulation is 1 second, which will be introduced in Section 5.1. We usually set training times *y* to 25. In order to make the total real time consumption 3∗f(x)∗y match the packet sending interval of NS-3 in simulation, the real time consumption of once HMM training for a single link can be calculated as 1/(25∗3)=0.0133 s. Therefore, the number of training points *x* is 10, according to Table 2.

The crucial criteria for evaluating the proposed HMM of the ANC scheme is to verify whether it can accurately predict the PLR in different vehicular network conditions. [35] proposed an estimation algorithm for PLR on VANET, which named RPLE. The core idea of RPLE is to use few probe packets and enhance the estimation accuracy on wireless networks. We evaluate the PLR estimation error range of the ANC scheme when compared with the RPLE algorithm. We use $Err_{i}^{PLR}$ to denote the estimation error. $Estimation_{i}$ means the *i* th estimation result and $PLR_{i}$ means the *i* th PLR point in validation set.
(15)$$Err_{i}^{PLR}=\frac{∥Estimation_{i}-PLR_{i}∥}{PLR_{i}}.$$

We collected PLR points in high-speed, low-speed, and static scenarios. Subsequently, we calculated the estimation error of three different methods in different mobile scenarios through a cumulative distribution function (CDF) graph. Figure 7 consists of three subfigures, in each subfigure, WDD+HMM represents the HMM estimation method with the average raw PLRs that are processed by WDD, WDD is introduced in the first paragraph of Section 4.4.2, WDD+HMM is the estimation method used in the ANC scheme. HMM represents the HMM estimation method without WDD processing. RPLE represents the estimation method that is used by [35]. In the high-speed scenario, 90% estimation errors of PLR for the WDD+HMM method are within 0.1, 75% estimation errors of PLR for the HMM method are within 0.1, and 70% estimation errors of PLR for the RPLE method are within 0.1, as shown in Figure 7a. All of the estimation errors of PLR for the WDD+HMM method are within 0.4, all estimation errors of PLR for the HMM method are within 0.7, and all estimation errors of PLR for the RPLE method are also within 0.7. Therefore, the estimation error range of PLR for the WDD+HMM is 30% smaller than that for the RPLE method and HMM method in the high-speed scenario. In the low-speed scenario, 95% estimation errors of PLR for the WDD+HMM method are within 0.1, 90% estimation errors of PLR for the HMM method are within 0.1, and 85% estimation errors of PLR for the RPLE method are within 0.1, as shown in Figure 7b. All of the estimation errors of PLR for the WDD+HMM method are within 0.4, all estimation errors of PLR for the HMM method are within 0.4, and all estimation errors of PLR for the RPLE method are within 0.5. Therefore, the estimation error range of PLR for ANC scheme is 10% smaller than that for the RPLE method in the low-speed scenario. Figure 7c shows that, in the static scenario, 99% estimation errors of PLR for the WDD+HMM method and HMM method are within 0.1 and 98% estimation errors of PLR for the RPLE method are within 0.1. All estimation errors of PLR for the WDD+HMM method are within 0.2 and all estimation errors of PLR for the HMM method and RPLE method are within 0.3. Therefore, the estimation error range of PLR for the ANC scheme is 10% smaller than that for RPLE and HMM in the static scenario. To sum up, the estimation results of the WDD+HMM method can filter the extreme shifts that are caused by the violent quality fluctuation of the vehicular wireless network on a small time scale to reduce the estimation error range of PLR in the high-speed scenario.

## 5. Results

In this section, we carry out a lot of simulations and real experiments in order to verify the superior of the ANC scheme compared with other NC schemes. First, we introduce our simulation setup in Section 5.1. Subsequently, in Section 5.2, we choose different link quality status in simulations to evaluate the throughput and transmission failure rate performance of the ANC scheme compared with the BNNC scheme, GFNC scheme, and OR scheme to verify the ability to combat the PLR variations of wireless links. Next, we verify the rationality of the simulation results through real experiments. We introduce our real experiment setup in Section 5.3. Finally, in Section 5.4, we evaluate the throughput and transmission failure rate performance of the ANC scheme compared with the BNNC scheme, GFNC scheme, and OR scheme in different real mobile scenarios in order to prove the rationality and validity of the simulation results.

### 5.1. Simulation Setup

For the simulations, we use Network Simulator 3 (NS-3) version 3.28 as the simulation platform, which is a discrete-event network simulator for Internet systems. NS3 is deployed in the Ubuntu 16.04 operating system and compiled with g++ 5.4.0. All of the NC schemes mentioned above were deployed in NS-3. In the simulation environment, there are a client and a server for sending and receiving data packets. The client simulates the MAR deployed in the vehicle and the server simulates the MER deployed in the cloud, the packet sending interval of the client in simulation is 1 second. To simulate the vehicle-to-cloud multipath transmission in vehicular IoT, we set three mutually independent wireless links between the client and server in NS-3.To simulate the cellular wireless links that are provided by existing 3 major ISPs in China, there are three mutually independent wireless links between the client and server in NS-3. Three links have the same bandwidth and delay configuration (bandwidth is 50 Mbps and delay is 15 ms). According to the different packet loss rate configuration of wireless links that are listed in Table 3, we simulate different wireless communication scenarios. Configuration 1 represents the static communication scenario, configuration 2 represents the low-speed communication scenario, and configuration 3 represents the high-speed communication scenario.

### 5.2. Network Performance of Different Network Coding Schemes in Simulations

The transmission failure rate is the ratio of the number of the coded groups that fail to be decoded to the total number of the coded groups. The transmission failure rate differs from PLR, because PLR regards a data packet as a statistical unit, but the transmission failure rate regards a coded group as a statistical unit. Suppose that there is only 1 group in transmission and this coded group consists of 10 coded data packets, and the coding rate of this coded group is 0.5, which means when more than 5 coded packets are lost in transmission, this coded group fails to be decoded. When 7 packets are lost in transmission, the transmission failure rate is 1, but the PLR is 0.3. TFR is an important metric for quantifying the ability of the NC scheme in order to overcome the packet loss problem of a wireless link in the different communication environments.

The simulation results are shown in Figure 8. Figure 8 consists of 3 subfigures, and each subfigure represents different communication scenario in simulations. Figure 8a shows the transmission failure rate and throughput performance of different NC schemes in a static communication environment. The transmission failure rates of all NC schemes are 0 in a static communication environment, but ANC has the highest throughput because ANC decreases its coding rate to reduce unnecessary bandwidth consumption resulted by redundant packets on the basis of overcoming the packet loss in transmission. The average throughput of ANC is close to 19.5 Mbps, the average throughput of BNNC is 18 Mbps, and the average throughput of GFNC is close to 17 Mbps. However, the average throughput of OR is close to 9 Mbps, which is much lower than these of other NC schemes, because OR has the most redundant packets, each time a data packet in one available wireless link is successfully received by the receiver, the remaining 2 redundant packets need to be discarded simultaneously in simulations. Figure 8b represents a low-speed communication environment, OR has the highest transmission failure rate compared with other schemes, because OR avoid packet loss by sending more replicated packets, OR actually can not recover lost packets. The transmission failure rate of ANC is still 0 but the average throughput of ANC is a little lower than that in the Figure 8a, this means ANC increase the number of redundant packets to overcome packet loss in transmission, more redundant packets consume more bandwidth, which results in relatively lower throughput. Figure 8c represents a high-speed communication environment, the throughput of the OR scheme is close to 5 Mbps, the throughput of ANC is close to 14 Mbps, the throughput of BNNC is close to 13 Mbps, and the throughput of BNNC is close to 11 Mbps. The throughput performance of all schemes degrades, because the PLRs of wireless links become higher and there is much more packet loss in transmission.

We put together the simulation data of a certain NC scheme in different scenarios in order to calculate the variance of transmission failure rate and the variance of throughput for different NC schemes. The variance of the simulation data gathered from different scenarios can reflect the fluctuation extent of throughput and transmission failure rate for different network coding schemes when the packet loss rate of wireless link changes. The lower the variance, the stronger the ability to counter the variations of PLR in wireless links. In order to make Figure 9a clearer, we use common logarithm as the ordinate in the Figure 9a, the abscissa in Figure 9 is the type of NC schemes. As shown in Figure 9, the throughput variance of ANC is 4, the transmission failure rate variance of ANC is close to $10^{-6}$. The ANC scheme has the lowest variances of transmission failure rate and throughput, which means that the ANC scheme has the strongest ability to counter the variations of PLR in wireless links when compared with other NC schemes for vehicle-to-cloud multipath transmission.

### 5.3. Real Experiment Setup

For the real experiments, we evaluate the throughput and transmission failure rate performance of the ANC scheme in the static environment and high-speed mobile environment. We choose the high-speed railway as the high-speed mobile environment. Along the Beijing-Guangdong express railway in China, we test the throughput and transmission failure rate performance of the ANC scheme, BNNC scheme, GFNC scheme, and OR scheme. We give the performance comparisons of different schemes in Section 5.4. Real experiment results verify the rationality and validity of the simulation results.

### 5.4. Network Performance of Different Network Coding Schemes in Real Experiments

The real experiment results are shown in Figure 10 and Figure 11. Figure 10 has two subfigures, Figure 10a is a comparison of the real-time throughput performance for different schemes in a static scenario. Figure 10b is a comparison of the real-time throughput performance for different schemes in a high-speed mobile scenario. For each subfigure in Figure 10, the abscissa represents the time(s), and the ordinate represents the throughput (Mbps). The black dotted line in each subfigure represents the general trend of network throughput for the ANC scheme. In a static scenario, ANC has the highest average network throughput, the throughput of the ANC scheme is close to 20 Mbps, as shown in Figure 10. The average throughputs of BNNC and GFNC schemes are close to 17 Mbps and 15 Mbps, respectively. The average throughput of the OR scheme is close to 12 Mbps because there is no packet loss in the static scenario, the throughput performance of the OR scheme is not affected by packet loss. In a high-speed mobile scenario, the ANC scheme also has the highest throughput, and the average throughput of the ANC scheme is close to 12.5 Mbps, as shown in Figure 10b. The average throughput of the BNNC scheme is close to 11 Mbps, the average throughput of the GFNC scheme is close to 10 Mbps, and the average throughput of the OR scheme is close to 5 Mbps. In terms of comparison with Figure 10a,b, we can find that the throughput performance of all schemes degrades in a high-speed scenario and the throughput degradation of the OR scheme is most obvious compared with other NC schemes. With the increase of moving speed, cellular wireless links are under the influence of Signals Fading and Doppler Effects, the PLR of the link becomes higher, there is much more packet loss in transmission. Therefore, the throughput performance of all NC schemes degrades.

Figure 11 has two subfigures, Figure 11b is a comparison of the average transmission failure rate in a static scenario and high-speed scenario for different schemes. Figure 11a is a comparison of the throughput variance for different schemes. We put together the real experiment data of a certain NC scheme from the static scenario and high-speed scenario in order to calculate the variance of throughput for different NC schemes to reflect the ability to counter the variations of link quality in wireless links. In Figure 11b, the average transmission failure rates of all NC scheme in the high-speed scenario is much higher than these of all NC schemes in the static scenario. We can find that OR has the highest average transmission failure rate when compared with other NC schemes in the high-speed scenario and static scenario. The transmission failure rate of the OR scheme also increases most obviously. This means that OR is easier affected by network fluctuations than other NC schemes. The throughput variance of the ANC scheme is the lowest compared with other NC schemes because the ANC scheme determines a reasonable network coding rate according to the estimated PLR to conquer the rapid changes of link quality, as shown in Figure 11a.

In terms of throughput and transmission failure rate performance, we can find real experiment results are similar to the simulation results, which also verifies the rationality of simulations and the usability of the ANC scheme.

## 6. Conclusions

In this paper, ANC is proposed in order to dynamically adjust the coding rate to counter the variations of link quality. We proposed a brand-new network coding scheme combined with HMM to significantly improve network throughput and reduce packet loss for vehicle-to-cloud multipath communication in vehicular IoT. Besides, the estimation part of the ANC scheme is a conjunction of wavelet and HMM to conquer the PLR variations in each available wireless link. When considering the differences between wireless links, the ANC scheme achieves the trade-off between the transmission reliability and the bandwidth cost based on the feedback of the estimated instantaneous PLR. In the end, extensive simulations and real tests are carried out along with comparisons with other NC schemes in order to verify the superiority of the ANC scheme in terms of throughput and transmission failure rate performance. The results show that ANC has the lowest transmission failure rate and achieves a smaller throughput degradation when network fluctuating. Under the background of the rapid development of vehicular IoT, the ANC scheme plays an important role in setting up a new method for improving the transmission quality of the cellular-based vehicular network. In our future work, we will focus on the research of combining multipath transmission with hierarchical edge computing in high-speed cellular-based vehicular network.

## Figures and Tables

**Figure 1 sensors-20-05902-f001:**
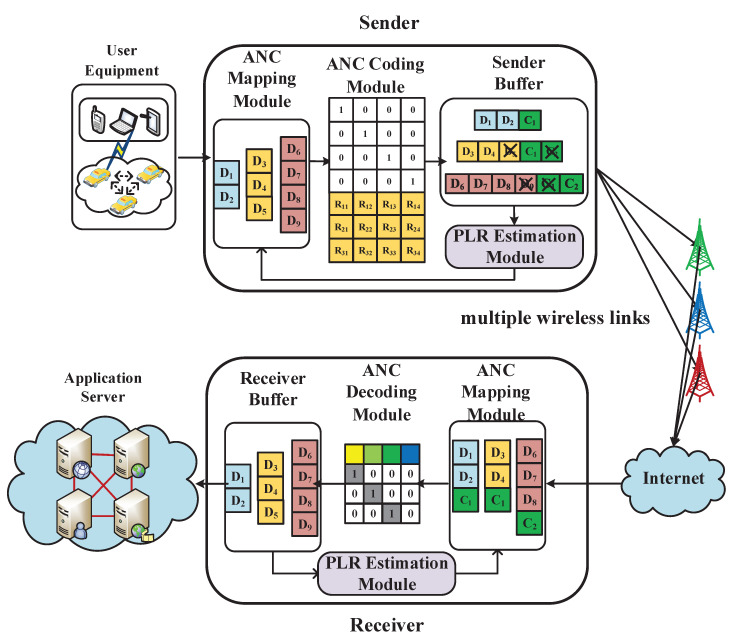
The architecture of the ANC scheme.

**Figure 2 sensors-20-05902-f002:**
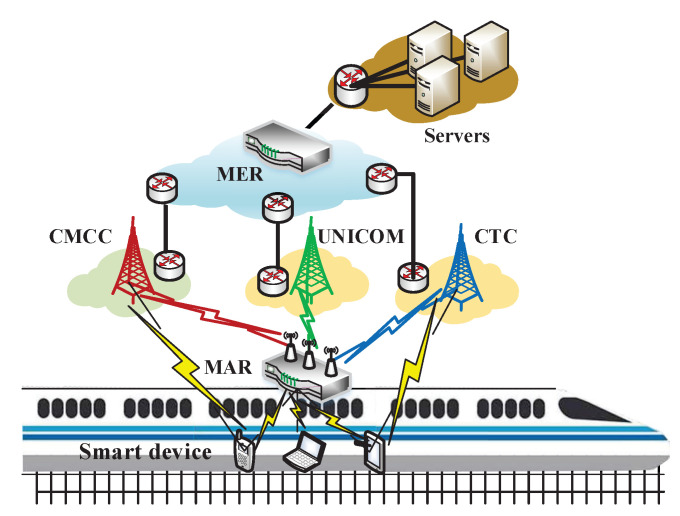
Vehicle-to-cloud multipath transmission network topology.

**Figure 3 sensors-20-05902-f003:**
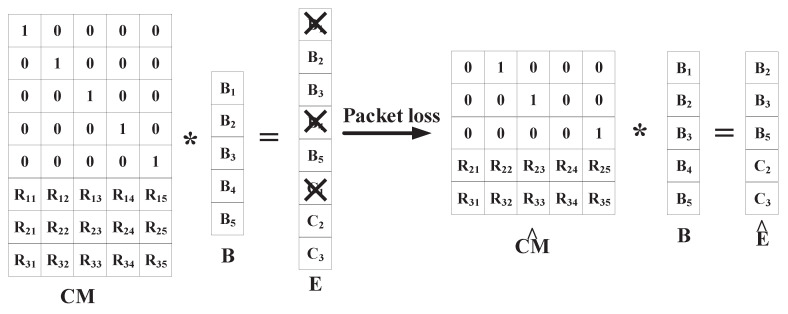
Coding process.

**Figure 4 sensors-20-05902-f004:**
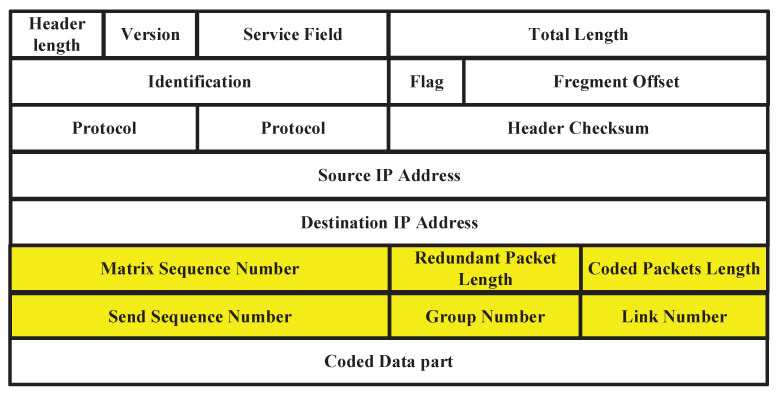
Customized header format in IP packet.

**Figure 5 sensors-20-05902-f005:**
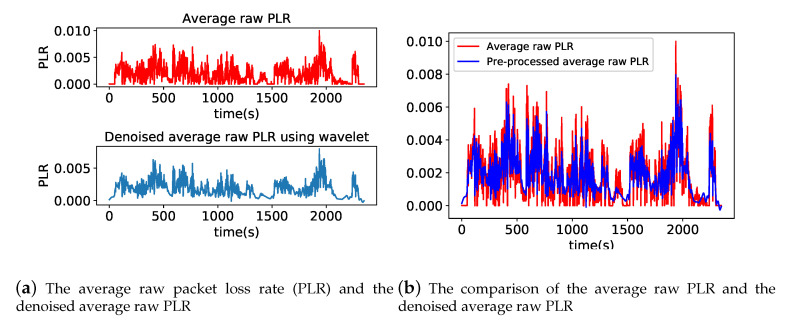
The wavelet domain de-noising process of average raw PLR.

**Figure 6 sensors-20-05902-f006:**
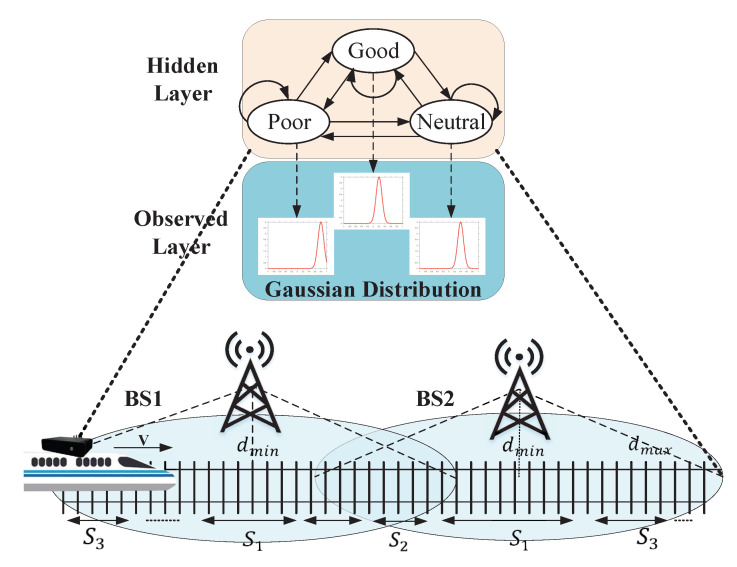
The estimation model of PLR.

**Figure 7 sensors-20-05902-f007:**
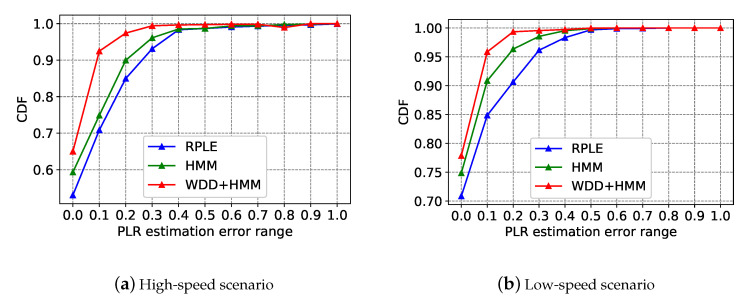
Comparison of estimation error range of PLR for different methods.

**Figure 8 sensors-20-05902-f008:**
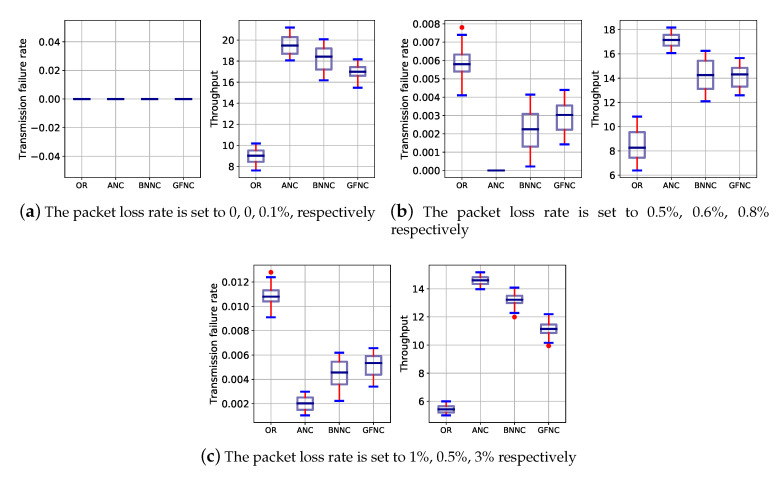
The transmission failure rate and throughput performance of different schemes in simulations.

**Figure 9 sensors-20-05902-f009:**
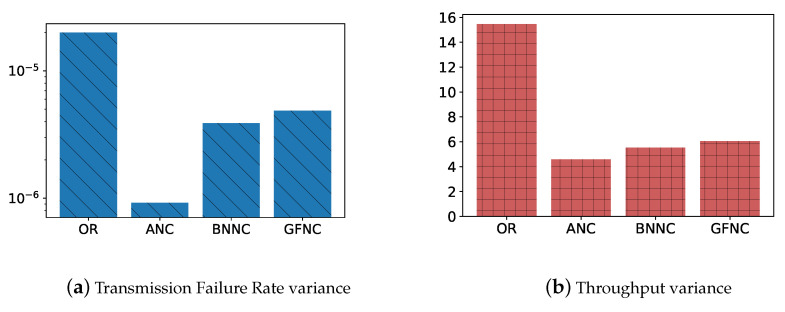
The transmission failure rate variances and throughput variances of different schemes in simulations. The variances reflect the fluctuation range of throughput and transmission failure rate for different network coding schemes when the packet loss rate of wireless link changes.

**Figure 10 sensors-20-05902-f010:**
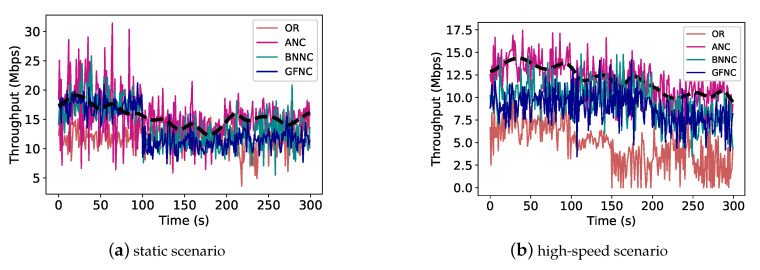
The real-time throughput performance of different schemes in high-speed and static scenarios: (**a**) The real-time throughput performance of different network coding schemes in the static scenario; (**b**) The real-time throughput performance of different schemes in the high-speed scenario.

**Figure 11 sensors-20-05902-f011:**
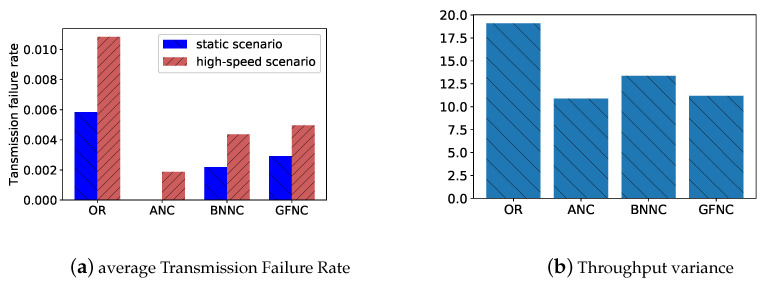
The average transmission failure rates and the throughput variances of different network coding schemes in real experiments. (**a**) the comparison of average transmission failure rates of each network coding scheme in the high-speed scenario and static scenario; and, (**b**) the variances of the combined data set of throughput in the high-speed scenario and static scenario, which reflect the fluctuation range of throughput for different network coding schemes when link quality changes.

**Table 1 sensors-20-05902-t001:** Abbreviation description.

Symbol	Description
**B**	Column vector of raw packets
**CM**	Coding matrix
**IM**	Identity matrix
**E**	Column vector of coded packets
$\hat{E}$	Column vector of received coded packets after transmission
$\hat{\mathrm{CM}}$	Reconstructed coding matrix in receiver
**D**	Decoding matrix
$k_{i}$	The length of raw packets group B in the *i* th link
$n_{i}$	The length of coded packets group E in the *i* th link
$r_{i}$	The number of row vectors of RM in the *i* th link
$p_{i}^{raw}$	Packet Loss Rate of the *i* th link
$BW_{i}$	Bandwidth of the *i* th link
$RTT_{i}$	Round-trip time of the *i* th link

**Table 2 sensors-20-05902-t002:** The real time consumption of once Hidden Markov Model (HMM) training with different points for a single link.

The Number of Training Points	Real Time Consumption for Single Link
10	0.01332179 s
11	0.01466731 s
100	0.02417990 s
110	0.02523301 s
1000	0.03267589 s
1100	0.03368711 s

**Table 3 sensors-20-05902-t003:** Different configurations of packet loss rate.

Configuration	Link A	Link B	Link C
1	0	0	0.1%
2	0.5%	0.6%	0.8%
3	1%	0.5%	3%

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
