# Peer review of "An Adaptive Network Coding Scheme for Multipath Transmission in Cellular-Based Vehicular Networks"

_sensors, 2020, doi:10.3390/s20205902_

Round 1
Reviewer 1 Report
The paper focuses on the quicker delivery of packets in the Vehicular IoT networks. It employs the Hidden Markov Model (HMM) into the Network Coding scheme to achieve greater throughput and reduce the packet loss in transmissions. The main contribution of the paper is the proposed algorithm to dynamically determine the proportion of coded packets allocated on each available wireless link.
The paper is well-written and well-organised, but the author could improve the quality further if they address the following comments:
Line 190: Vehicle-to-could multipath communication needs a proper definition or at least an elaborate explanation to clarify what it means to the readers.
Line 209: Equation 1 ignores the processing delay and queuing delay in transmission. However, in wireless networks, queueing delay could become a challenge if the number transmissions (or vehicles) increases significantly. The author may consider adding an explanation as to how they think their scheme would react in response to such events or why such a queuing delay will not affect the scheme at all.
Line 302: The author rightly identified that the network quality fluctuations have been more fierce in the high-speed scenario. As a mitigating approach, they used Daubechies8 wavelets to process the average raw PLRs. However, I feel a justification was necessary defending authors' choice of method and how it helps in this problem.
Line 356: Ten observed raw PLR points used in training the model. Why ten and why not more— an explanation would strengthen the discussion.
Line 395: The author presented both simulated and real experimental results. Although I understood the need for two versions of experiments, it was the authors' responsibility to clarify this at the beginning of the simulation section. It could easily make readers confused as they think in the presence of real experiments, what's the point of having a set of extra simulated experiments.
Reviewer 2 Report
The authors propose an Adaptive Network Coding (ANC) scheme brought by the novel integration of the Hidden Markov Model (HMM) into the NC scheme to efficiently adjust the coding rate according to the estimated packet loss rate (PLR). The ANC scheme conquers the rapid change of wireless link quality to get the utmost throughput and reduces the packet loss in transmission. Their simulation and real experiment results show the throughput performance of the ANC scheme outperform state-of-the-art NC schemes for vehicular wireless multi-path transmission in vehicular IoT systems.
The paper is very well written and is scientifically sound, bringing a contribution to the field of Vehicular Networks.
However, there are some minor issues that need to be addressed by the authors before the paper can be published:
1- Increase the font size in figures 1, 5, and 6;
2- The authors simulate three wireless networks ("... are provided by existing 3 major ISPs in China..."). My questions are:
How the simulation of the characteristics of these three wireless networks in China would impact the results obtained? Is it possible to extrapolate these results to be used elsewhere in the world? What are the impacts (if any) of these considerations?
